# Identification and characterization of miRNAs involved in cold acclimation of zebrafish ZF4 cells

**Xiangqin Ji[1,2], Penglei Jiang[1,2], Juntao Luo[1,2], Mengjia Li[1,2], Yajing Bai[1,2], Junfang Zhang[1,3], Bingshe Han[1,2]***

**1** Key Laboratory of Exploration and Utilization of Aquatic Genetic Resources, Shanghai Ocean University, Ministry of Education, Shanghai, China, **2** National Demonstration Center for Experimental Fisheries Science Education, Shanghai Ocean University, Shanghai, China, **3** International Research Center for Marine Biosciences at Shanghai Ocean University, Ministry of Science and Technology, Shanghai, China

* bs-han@shou.edu.cn

**Data Availability Statement:** The data used in this study has been deposited in NCBI's Gene Expression Omnibus repository and are accessible through GEO accession number GSE134399.

## Abstract

MicroRNAs (miRNAs) play vital roles in various biological processes under multiple stress conditions by leading to mRNA cleavage or translational repression. However, the detailed roles of miRNAs in cold acclimation in fish are still unclear. In the present study, high-throughput sequencing was performed to identify miRNAs from 6 small RNA libraries from the zebrafish embryonic fibroblast ZF4 cells under control (28°C, 30 days) and cold-acclimation (18°C, 30 days) conditions. A total of 414 miRNAs, 349 known and 65 novel, were identified. Among those miRNAs, 24 (19 known and 5 novel) were up-regulated, and 23 (9 known and 14 novel) were down-regulated in cold acclimated cells. The Gene Ontology (GO) and Kyoto Encyclopaedia of Genes and Genomes (KEGG) enrichment analyses indicated that the target genes of known differentially expressed miRNAs (DE-miRNA) are involved in cold acclimation by regulation of phosphorylation, cell junction, intracellular signal transduction, ECM-receptor interaction and so on. Moreover, both miR-100-3p inhibitor and miR-16b mimics could protect ZF4 cells under cold stress, indicating the involvement of miRNA in cold acclimation. Further study showed that miR-100-3p and miR-16b could regulate inversely the expression of their target gene (*atad5a*, *cyp2ae1*, *lamp1*, *rilp*, *atxn7*, *tnika*, *btbd9*), and that overexpression of miR-100-3p disturbed the early embryonic development of zebrafish. In summary, the present data show that miRNAs are closely involved in cold acclimation in zebrafish ZF4 cells and provide information for further understanding of the roles of miRNAs in cold acclimation in fish.

## Introduction

MicroRNAs (miRNAs) are a class of endogenous non-coding RNAs in plants and animals, approximately 22 nucleotides in length, which play key roles in regulating gene expression by leading to cleavage or translational repression of their target mRNA [1, 2]. Usually a single miRNA may target up to hundreds of protein-coding genes, and it is estimated that 74% to

**Funding:** This study was supported by grants from National Natural Science Foundation of China (grant No. 81770165. awarded to BH and 31372516 awarded to JZ).

**Competing interests:** The authors have declared that no competing interests exist.

**Abbreviations:** miRNA, microRNA; sRNA, small RNA; ZF4, zebrafish embryonic fibroblast cells; GO, Gene Ontology; KEGG, Kyoto Encyclopaedia of Genes and Genomes; TPM, Transcripts per million; DE, Differentially Expressed.

92% of all protein-coding genes are conserved miRNA targets [3–5]. Many miRNAs are evolutionarily conserved and involved in growth, development, apoptosis, proliferation, and tumorigenesis of organisms [1, 6].

Temperature is a major factor that affects the life of fish, which are especially sensitive to temperature conditions during early development, and influences multiple biological processes of aquatic organisms including behavior, development, sex determination and so on [7, 8]. Increasing reports have shown that specific protein-coding genes are differentially expressed under cold stress [9, 10], and miRNAs play crucial roles in this process [11–13]. A number of studies have demonstrated that miRNAs have significant roles in response to cold stress in numerous species, such as arabidopsis [14], wheat [15], rat [16], *Litopenaeus vannamei* [17], and zebrafish [18]. However, most studies focus on plants and mammals under acute cold stress, the regulatory mechanisms of miRNA in long-term cold acclimation in fish are still obscure.

Zebrafish (*Danio rerio*), a fresh water tropical cyprinid fish with short breed cycle and big spawn amount, is one of the most important vertebrate models in genetics, development, and biomedicine [19, 20]. Zebrafish is also a good model to study cold acclimation with a wide temperature range of approximately 16–40˚C [8]. Our previous research showed that DNA methylation and long non-coding RNAs (lncRNAs) are involved cold acclimation of zebrafish ZF4 cells when experimentally acclimated at 18˚C for 30 days [21, 22].

In this study, we investigated the roles of miRNAs in cold acclimation of ZF4 cells for a better understanding of the epigenetic mechanisms involved in cold acclimation in fish. We identified and characterized the miRNAs responding to cold acclimation and predicted the functions of those miRNAs. And we showed that dre-miR-16b and dre-mir-100-3p contribute to cold acclimation by regulation of cell survival under cold stress. Our data will contribute to a full understanding of the mechanisms of miRNA in fish under cold pressure.

## Materials and methods

### Cell culture and treatment

ZF4 cell line was from the American Type Culture Collection (ATCC, CRL 2050). Cells were cultured in Dulbecco's modified Eagle's medium/F12 nutrient mix (SH30023.01B, Hyclone, Thermo Scientific) supplemented with 10% fetal bovine serum (10099141, Gibco, Life technologies), 1% penicillin-streptomycin-glutamine solution (SV30082.01, Hyclone, Thermo Scientific) at 28˚C, 5% $CO_2$. For cold acclimation, ZF4 cells were seeded at 50% confluence and the next day transferred to an incubator at 18˚C, 5% $CO_2$, in the same medium for 30 days. For recovery treatment, cold acclimated ZF4 cells were transferred to an incubator at 28˚C, 5% $CO_2$ for up to 10 days.

### Small RNA library construction and sequencing

Total RNA was extracted using miRNeasy Mini Kit (217004, Qiagen), and purified by RNAClean XP Kit (A63987, Beckman Coulter) and RNase-Free DNase Set (79254, Qiagen) according to the manufacturer's instructions. The quality and integrity of RNAs were examined using an Agilent 2100 Bioanalyzer (Agilent technologies, Santa Clara, US). Libraries were constructed using TruSeq Small RNA Sample Prep Kit (RS-200-0012/ RS-200-0024, Illumina). Libraries were pooled and sequenced using the Illumina HiSeq machine as 50-bp single-end sequencing reads. Libraries were constructed from 3 control and 3 cold acclimation samples.

## Identification of known and novel miRNAs

Clean reads were obtained from raw sequences after removing all low-quality reads (N% > 5%, or with ploy A), contaminants, reads smaller than 17 nt or longer than 35 nt, and trimming the adapters. The clean reads were used to calculate length distribution and base preference. Moreover, clean reads mapped to the zebrafish non-coding RNA sequences (danRer10, ftp://ftp.ensembl.org/pub/release-91/fasta/danio_rerio/ncrna/danio_rerio/ncrna/) using bowtie 2.3.0 [23] were annotated as microRNA (miRNAs), transfer RNA (tRNAs), ribosomal RNA (rRNAs), small nuclear RNA (snRNA) or other small RNAs. To identify the miRNAs associated with cold acclimation, clean reads were aligned to the known zebrafish miRNA database miRBase22 (http://www.mirbase.org/) using bowtie 2.3.0. Novel miRNAs and their precursor structures were predicted by miRCat [24]. According to the folding model, sequences locate in the stem-loop structure with mean counts greater than 20 and expression in all samples, were defined as candidate novel miRNAs.

## Identification of differentially expressed miRNAs (DE-miRNAs)

The Deseq R package [25] was used to analyze DE-miRNAs during cold acclimation. DE-miRNAs were defined as the ones with fold change (FC) >2.5 and adjusted p-value < 0.01 and a mean expression level greater than 20.

## Target prediction and functional annotation of miRNAs

Target genes of miRNAs were predicted using the miRanda software [26]. The miRanda Score ≥ 150 and Energy ≤ -24 kcal/mol were used to select unigene targets. We used Database for Annotation, Visualization and Integrated Discovery (DAVID) v6.8 web tool (https://david.ncifcrf.gov/) to perform GO and KEGG enrichment analyses with a significance of P < 0.05 [27, 28].

## Quantification of miRNA and mRNA by qRT-PCR

For miRNA assay, small RNA was extracted using SanPrep Column microRNA Extraction Kit (B518811, Sangon Biotech, China), and reverse transcription (RT) was performed using 200 ng of RNA with miRNA First Strand cDNA Synthesis (Tailing Reaction) (B532451, Sangon Biotech, China). The forward primers for miRNAs were designed and synthesized by Sangon Biotech (Shanghai, China), and the reverse primers are the universal set provided by Sangon Biotech. U6 was used as the internal control. For mRNA assay, total RNA was extracted using TRlzol reagent (15596–026, Life Technologies), and reverse transcription was performed using 1 μg of total RNA with PrimeScript™RT reagent Kit with gDNA Eraser (RR047A, TaKaRa), according to the manufacturer's instructions. β-actin was used as the internal control. Thereafter, qRT-PCR was performed using a total reaction volume of 10 μL, which contained 1 μL of diluted cDNA, 10 μM primer mix, 5 μL of 2× SYBR Green Mix, and 3 μL of ddH$_2$O. The reactions were carried out on a CFX96 Real Time PCR System (Bio-Rad, USA) with the following amplification conditions: 5 min at 95˚C, followed by 40 cycles at 95˚C for 10 s and 58˚C for 20 s, and a melt curve generated from 65˚C to 95˚C. Relative changes were analyzed using the $2^{-\Delta\Delta CT}$ relative quantification method [29]. Statistical analysis was performed using GraphPad Prism 5 software. The Student's T test was used for measurements of gene expression of samples from control group and cold acclimation group, and 3 experimental replicates were prepared for each condition. Primers for qRT-PCR analysis are shown in S1 Table.

## miRNA transfection and cell viability assay

About $3\times10^5$ ZF4 cells were seeded into each well of a 6-well plate. The next day, dre-miR-100-3p mimic (100 pmol), dre-miR-16b mimic (100 pmol), dre-miR-100-3p inhibitor (300 pmol), dre-miR-16b inhibitor (300 pmol), mimics control (100 pmol) or inhibitor control (300 pmol) (GenePharma, China) was introduced into ZF4 cells respectively using Attractene Transfection Reagent (301005, Qiagen) according to the manufacturer's instructions. The sequences of above mimics and inhibitors are shown in S1 Table. One day later, ZF4 cells were moved into an incubator at 10˚C, 5% $CO_2$ for cold treatment. After 36 hours, cells were prepared into a cell suspension and stained with 0.4% Trypan Blue solution (w/v) for 5 min, then counted with a hemocytometer.

## Fish husbandry and microinjection

The experimental protocol was approved by the Animal Ethics committee of Shanghai Ocean University and abides by the Guidelines on Ethical Treatment of Experimental Animals established by the Ministry of Science and Technology, China. Zebrafish were maintained and staged according to standard methods. The wild-type (AB) zebrafish was used for all experiments. Breeding fish were maintained at 28˚C in a circulating water system on a 14-h light/10-h dark cycle. The developing embryos were kept in an incubator at constant temperatures. dre-miR-100-3p mimics (5uM), dre-miR-100-3p inhibitor (20uM), mimics control (5uM), or inhibitor control (20uM) (GenePharma, China) was injected into embryos at one-cell stage (1 nL per embryo) [30].

# Results

## High-throughput sequencing of sRNAs

Zebrafish ZF4 cells were maintained at 18˚C for 30 days for cold acclimation [22], then 3 RNA samples from control group (con 1,2,3) and 3 RNA samples from cold acclimation group (cold 1,2,3) were subjected to sequencing. An overview of the workflow to process raw data and predict miRNAs and their targets is shown in Fig 1. The numbers of clean reads are 15,433,630 (con 1), 15,138,670 (con 2), 16,554,364 (con 3), 17,537,374 (cold 1), 16,141,740 (cold 2), and 17,067,851 (cold 3) respectively. Reads without sRNA sequences, ranging from 18 to 35 nt in length, were filtered using cutadapt (Fig 2A). The majority of the RNA sequences range from 21 to 24 nt in length, conforming to the characteristic length distribution pattern of miRNA libraries (Fig 2A). In terms of the 5′ nt, the majority of the miRNA start with U, which is the characteristic base distribution pattern of miRNA libraries (Fig 2B) [31]. To classify the sequenced reads and identify miRNA sequences in the 6 libraries, reads were mapped to the ncRNA database (Fig 2C). As a result, 88.60% and 86.69% sRNA reads were perfectly mapped to the zebrafish ncRNA database from control and cold acclimation libraries, respectively.

## Identification of known and novel miRNAs

The clean reads were aligned with known miRNAs in miRBase, and totally 349 known miRNAs were obtained (S2 Table). There are 339 and 341 known miRNAs identified for control and cold acclimation conditions, respectively (Fig 3B). The expression of know miRNAs in each library was statistically analyzed and normalized using TPM (Transcripts per million). TPM analysis showed that miRNAs underwent increased expression during cold acclimation (Fig 2A). Reads that failed to be aligned to the ncRNA database were used to predict novel miRNAs, and 65 novel miRNAs were obtained after identification and prediction of precursor structures using miRCat. (S3 Table). All novel miRNA candidates were selected with the

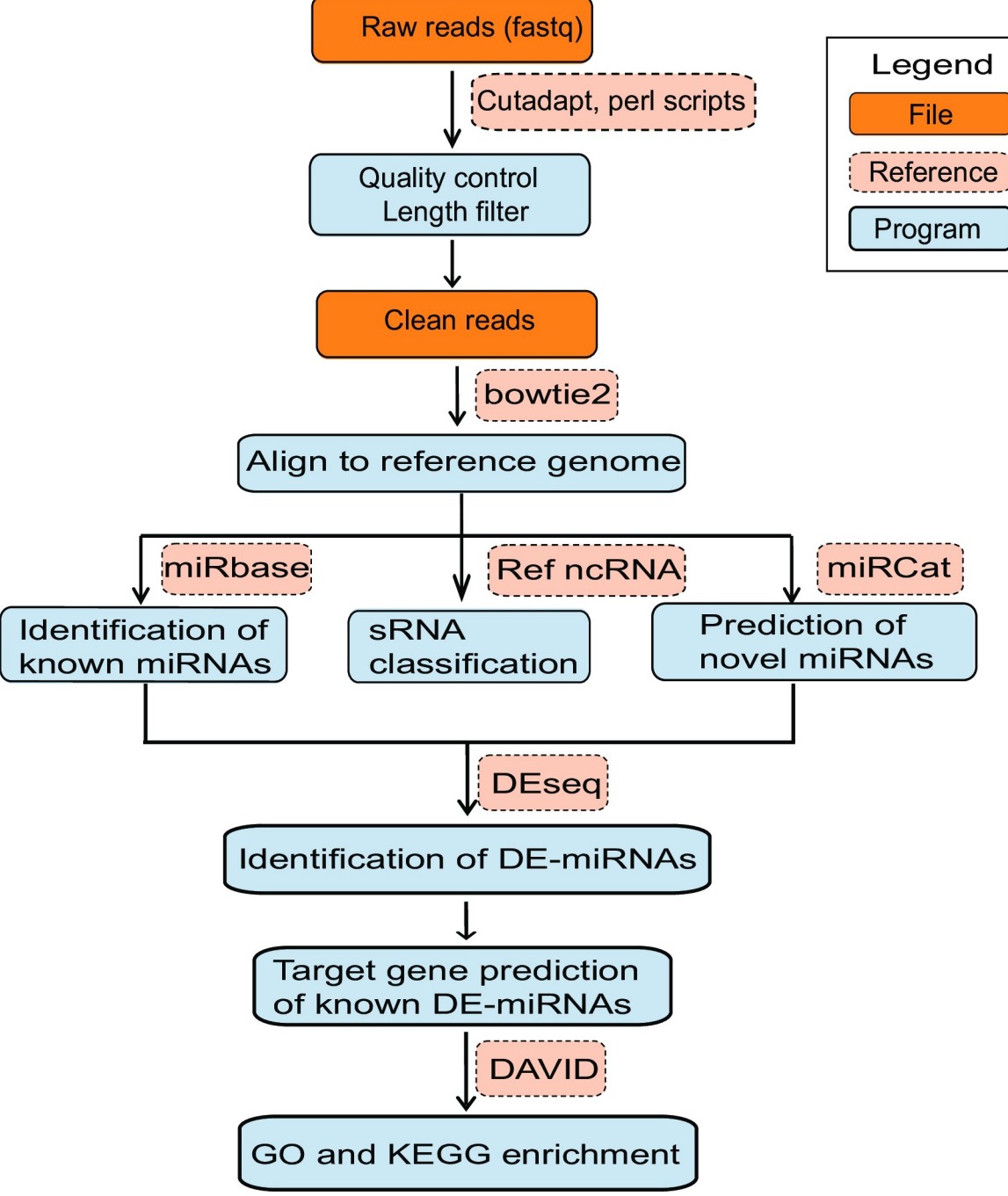

**Fig 1. Pipeline workflow of small RNA search and target prediction.** miRBase: microRNAs database, Ref ncRNA: reference non-coding RNA sequences, miRNAs: microRNAs, sRNAs: small RNAs, DE-miRNAs: differentially expressed miRNAs, DAVID: Database for Annotation, Visualization and Integrated Discovery, GO: Gene Ontology, KEGG: Kyoto Encyclopaedia of Genes and Genomes.

default parameters. The criteria for novel miRNAs are minimum free energy <-18 kcal/mol, abundance >20, and expression in all samples. The mature sequences of all identified miRNAs are 19–24 nt in length. The secondary structures of those novel miRNA are summarized in S4 Table.

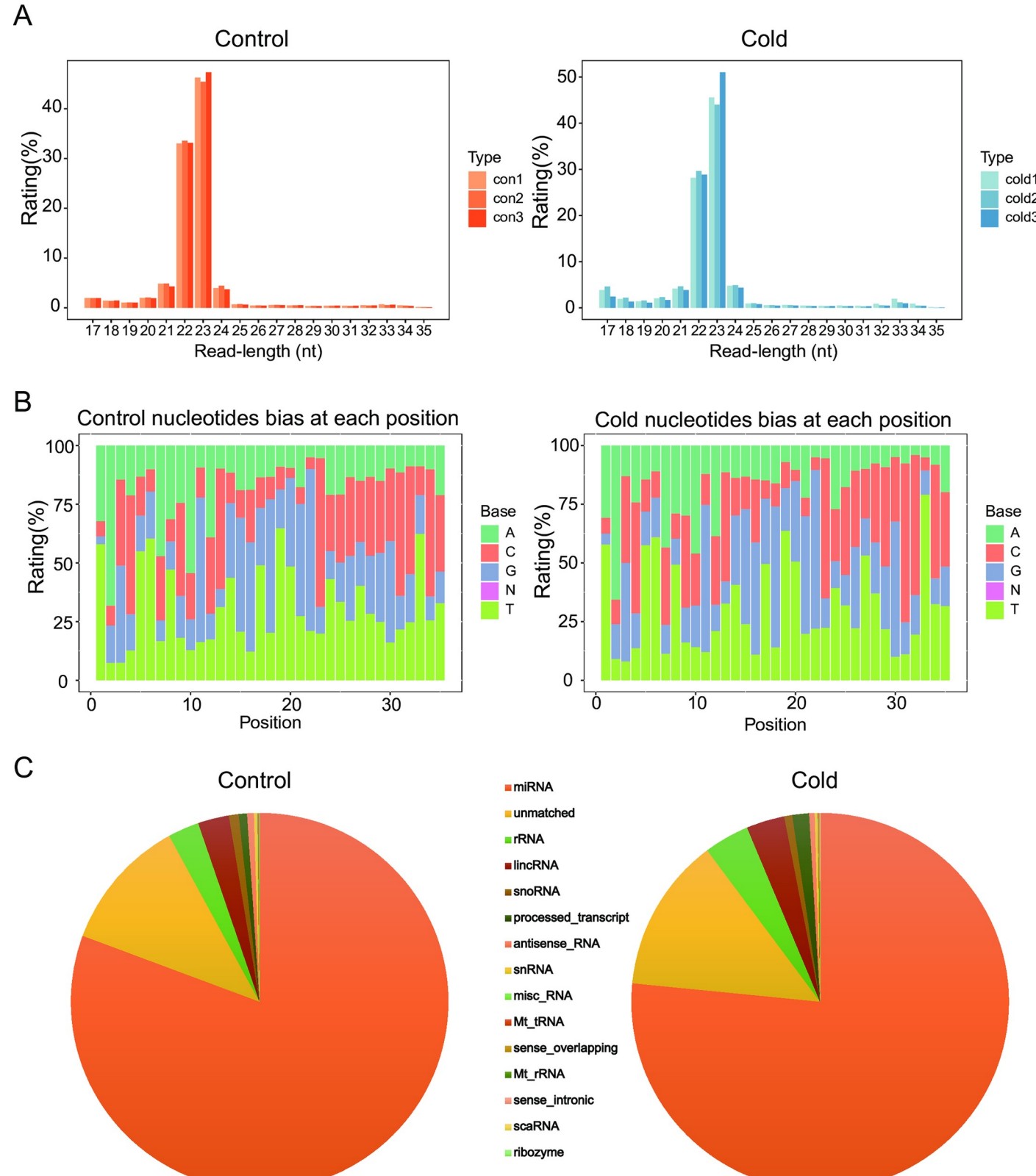

**Fig 2. Statistics and classification of small RNAs.** Read length distribution of small RNA libraries (A) and nucleotides bias at each position of miRNA (B) for control or cold acclimation condition. (C) An overview of the frequency of different RNA species present in libraries from each group. miRNA: microRNA,

unmatched: unmatched reads, rRNA: ribosomal RNA, lincRNA: long intergenic non-coding RNA, snoRNA: small nucleolar RNA, snRNA: small nuclear RNA, misc_RNA: miscellaneous RNA, Mt_tRNA: mitochondrial transfer RNA, Mt_rRNA: mitochondrial ribosomal RNA, scaRNA: small cajal body-specific RNA.

## Differentially expressed miRNA (DE-miRNAs) during cold acclimation

Next we investigated the miRNAs involved in cold acclimation. Considering that low-expressed sequences have more noise than high-expressed sequences, reads with expression level greater than 20 were used to identify DE-miRNAs [32]. Differential expression analysis revealed 24 (19 known and 5 novel) up-regulated and 23 (9 known and 14 novel) down-regulated miRNAs (Fold Change ≥ 2.5, adjusted p-value ≤ 0.01) by DEseq (S5 Table). More information of DE-miRNAs is provided in S5 Table. Cluster analysis of known DE-miRNAs (Fig 3C) and cluster analysis of novel DE-miRNAs (S1 Fig) were performed to show the expression patterns of those miRNA during cold acclimation, the novel miRNAs showed more significant variation in cold acclimation compare with the known miRNAs. Then the high-throughput sequencing data were validated by qRT-PCR, and all 8 miRNA selected showed the same tendency in sequencing and qRT-PCR (Fig 3D). Meanwhile, the cold acclimated cells were moved back to 28˚C and cultured for another 10 days, then those 8 miRNAs were detected using qRT-PCR. We noticed a recovery of the expression of all tested miRNAs with different extents, indicating that the response of those cold related miRNA disappeared after cold pressure was removed (Fig 3D).

## Target prediction and functional analysis of DE-miRNAs

A total of 3,627 target genes for 28 known DE-miRNAs were predicted using the miRanda software (S6 Table). GO and KEGG enrichment analyses of the differentially expressed protein-coding genes were also performed (S7 Table). GO enrichment analysis revealed 44 biological processes, including phosphorylation, methylation, chordate embryonic development, transmembrane transport, regulation of autophagy, and so on. The top 10 (ranked according to p-value) biological processes are shown in Fig 4A. And enriched KEGG pathways include focal adhesion, FoxO signaling pathway, adherens junction, lysine degradation, dorso-ventral axis formation, ECM-receptor interaction, and ABC transporters (Fig 4B). Those results indicated that miRNAs participate in cold acclimation of zebrafish via regulation of above biological processes.

The effect of miR-16b and miR-100-3p on their predicted target genes was analyzed. miR-100-3p mimics, miR-16b mimics, miR-100-3p inhibitor, or miR-16b inhibitor was transfected into ZF4 cells respectively, then the mRNA levels of selected target gene were examined using qRT-PCR. An inverse expression pattern was observed among the miRNAs and their predicted target genes (*atad5a*, *cyp2ae1*, *lamp1* targeted by miR-100-3p, and *rilp*, *atxn7*, *tnika*, *btbd9* targeted by miR-16b) (Fig 5). The binding sites between the miRNAs and their target genes are show in S2 Fig.

## Roles of miR-100-3p and miR-16b during cold acclimation

Since miR-16b and miR-100-3p are closely related to cell proliferation and apoptosis [33–35], the roles of miR-100-3p and miR-16b during cold acclimation were investigated. ZF4 cells were transfected with miR-100-3p mimics, miR-16b mimics, miR-100-3p inhibitor or miR-16b inhibitor, respectively, and incubated at 10˚C as cold treatment, then cell viability was detected. miR-100-3p inhibitor played a protective role in ZF4 cells under cold stress while miR-100-3p mimics decreased cell survival (Fig 6A), indicating that down-regulation of miR-100-3p (Fig 3C) is helpful for cell survival under cold stress. Meantime, miR-16b mimics

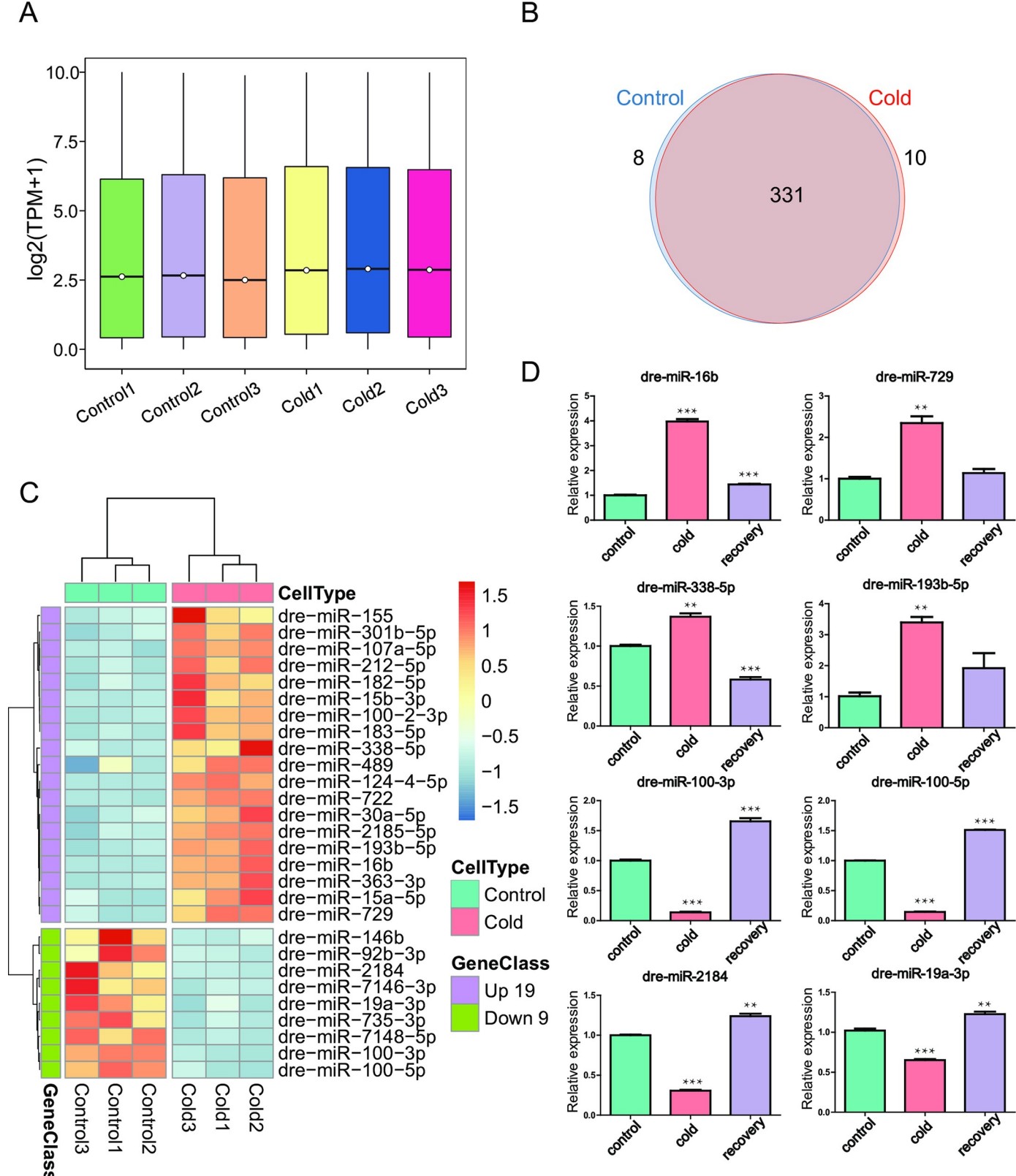

**Fig 3. Identification and annotation of small RNAs.** (A) Comparison of miRNA expression levels between small RNA libraries from control and cold acclimated ZF4 cells. (B) Venn diagram shows the number of known miRNAs expressed only in control and cold acclimated ZF4 cells. (C) A heatmap was generated based on

fold change values of known DE-miRNAs to visualize the expression patterns of the cold responsive miRNAs. (D) ZF4 cells were cultured at 18˚C for 30 days, then returned to 28˚C and cultured for another 10 days for recovery. The expression of indicated miRNAs was detected using qRT-PCR. The data are presented as means ± SD of three independent replicates. $p < 0.05$ was considered to indicate a statistically significant result. \*: $p < 0.05$, \*\*: $p < 0.01$, \*\*\*: $p < 0.001$.

enhanced cell survival under cold stress when miR-16b inhibitor led to decreased cell viability (Fig 6B), indicating that up-regulation of miR-16b can protect cells under cold stress. Above data suggested that miR-100-3p and miR-16b play important roles in ZF4 cells during cold acclimation via modulating cell survival.

The role of miR-100-3p in zebrafish embryonic development was further investigated by microinjection of miR-100-3p mimics or inhibitor. qRT-PCR showed that miR-100-3p level was significantly decreased or increased by its inhibitor or mimics (Fig 6C). From 6 hpf, miR-100-3p mimics injected embryos showed an increased mortality rate, and no significant change was observed in embryonic mortality in miR-100-3p inhibitor injected embryos. Our results suggested that miR-100-3p overexpression affects the early development of zebrafish embryos.

## Discussion

Increasing evidence supports the involvement of miRNAs in cold stress of plants and animals. miR408 and its target genes show regulatory roles in cold response in arabidopsis [14]. miR-319 is a potential marker for selection of cold-tolerant sugarcane cultivars [36]. miR-210-3p modulates expression of genes related to metabolism, apoptosis and proliferation in rat cells under acute cold stress conditions [16], and miRNAs play key roles in cold adaptation of *Litopenaeus vannamei* [17]. Although mechanisms related to genome, transcriptome, DNA methylation, histone modification and so on have been reported in cold responses of fish [21, 37–40], the detailed mechanisms of miRNAs during cold acclimation in fish are still unclear.

In the present work, we investigated the variation of miRNA expression in cold acclimated ZF4 cells, which were maintained at 18˚C for 30 days, through high-throughput sequencing. The expression of numerous miRNAs significantly altered in cold acclimated cells, including dre-miR-16b, dre-miR-2185-5p, dre-miR-100-3p, dre-miR-100-5p, dre-miR-19a-3p, dre-miR-7148-5p and other 21 known miRNA. The sequencing data were validated with 8 selected miRNAs, which all showed a tendency of recovery after cold stress was removed, indicating the cold response of miRNA is reversible.

Furthermore, 3,627 DE-mRNAs were predicted as target genes of 28 DE-miRNAs. The DE-mRNAs were enriched in biological processes like regulation of GTPase activity, cell adhesion, phosphorylation, membrane disruption in other organism, and KEGG pathways like ECM-receptor interaction, FoxO signaling pathway, focal adhesion, adherens junction. Most of enriched biological processes and pathways are consistent with previous researches about cold adaptation in fish [22, 41, 42], indicating that those putative cold-related processes and pathways are under regulation of multiple levels of genetic, molecular, and epigenetic mechanisms.

Next we investigated the roles of selected DE-miRNAs in cold acclimation. Significant down-regulation of dre-miR-100-3p indicated that this miRNA may play important roles in cold acclimation. Our data with its mimics or inhibitors showed that dre-miR-100-3p inhibitor and dre-miR-16b mimics could protect ZF4 cell under cold stress, suggesting that miR-100-3p participate in cold acclimation via supporting cell viability. It has been reported that miR-100 significantly inhibits the migration and invasion of NPC cells by targeting IGF1R [43], miR-100 enhances cisplatin induced autophagy and apoptosis in MG-63/DDP cells via targeting mTOR [33]. Here we showed that *atad5a*, *cyp2ae1* and *lamp1* have a putative binding site of

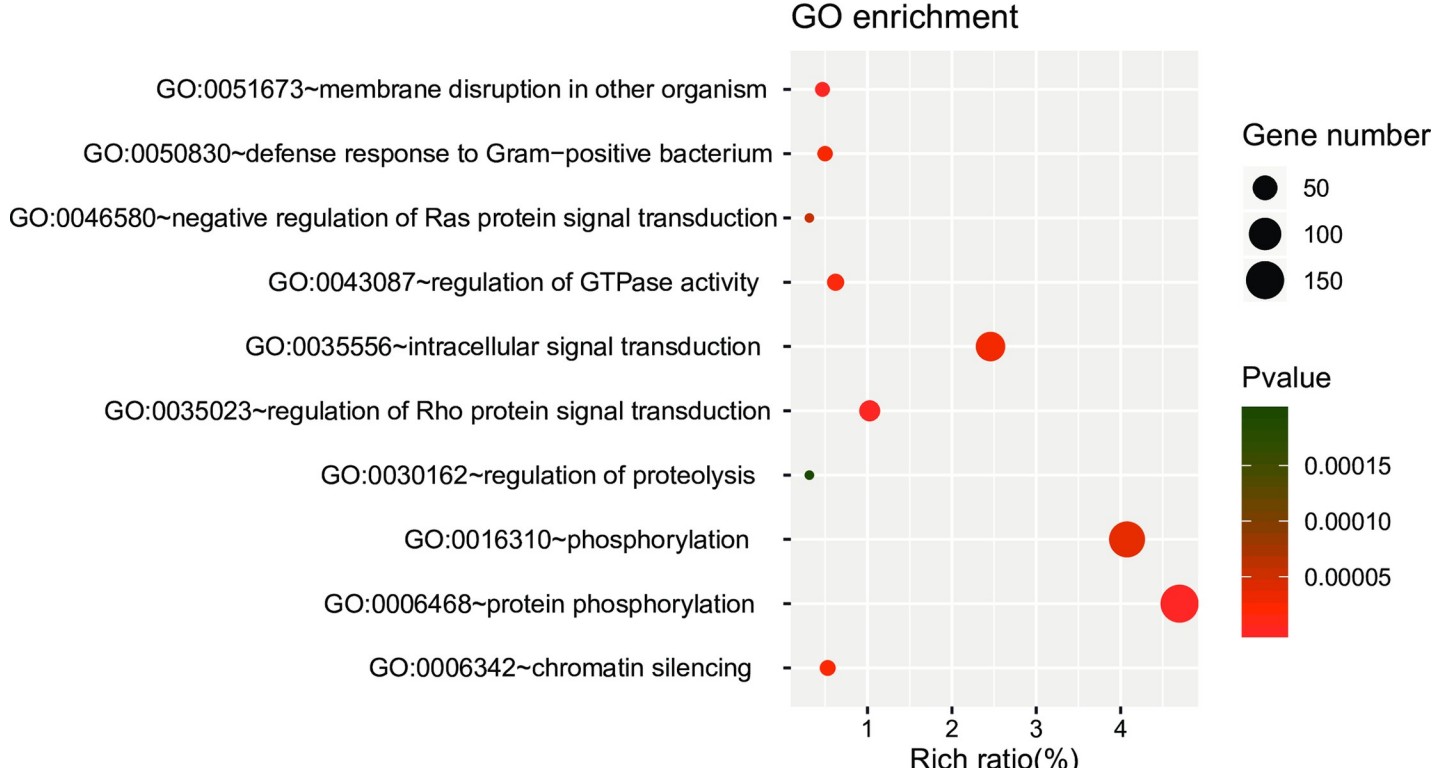

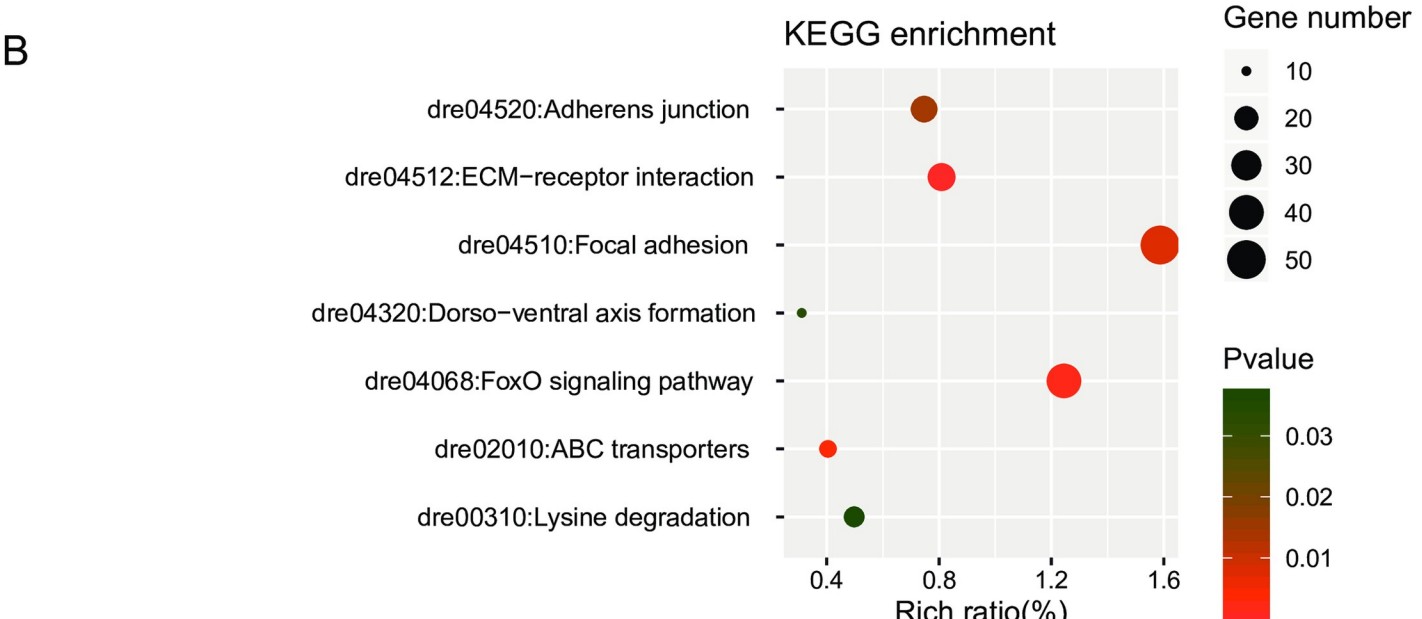

**Fig 4. Identification and functional prediction of DE-miRNAs.** (A-B) GO and KEGG enrichment analyses of the target genes of DE miRNAs. The y-axis corresponds to GO/ KEGG pathway with a p-value ≤ 0.05, and the x-axis shows the enrichment ratio between the number of DE-genes and all unigenes enriched in a particular pathway. The color of the dot represents p-value, and the size of the dot represents the number of DE-genes mapped to the reference pathways.

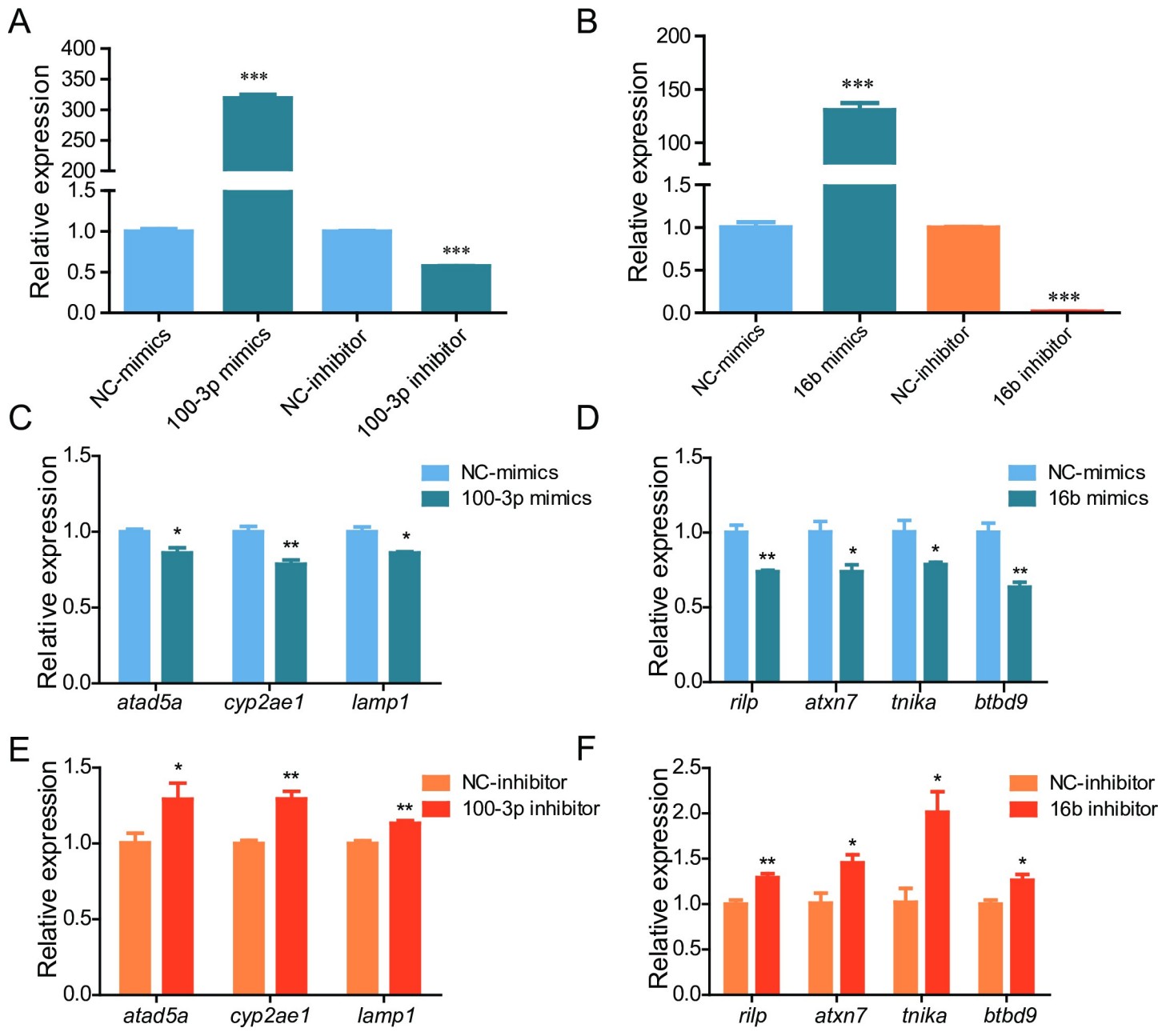

**Fig 5. Effect of selected miRNAs on their target genes.** ZF4 cells were transfected with indicated mimics or inhibitor, 24 hours later the expression of miR-100-3p or miR-16b (A-B) and their target genes (C-F) was detected by qRT-PCR. NC: negative control. *: p <0.05, **: p <0.01, ***: p <0.001.

miR-100-3p, respectively, and an inverse expression pattern was observed between these genes and miR-100-3p, indicating these genes may contribute to miR-100-3p function by regulating cell adhesion, metabolic reactions and ATP synthesis [44, 45]. The possibility that dre-miR-100-3p contributes to cold acclimation via above mentioned mechanisms needs to be investigated.

dre-miR-16b was up-regulated in cold acclimation in the present study, which is consistent with the report that the expression of miR-16 increases in platelets under cold-storage

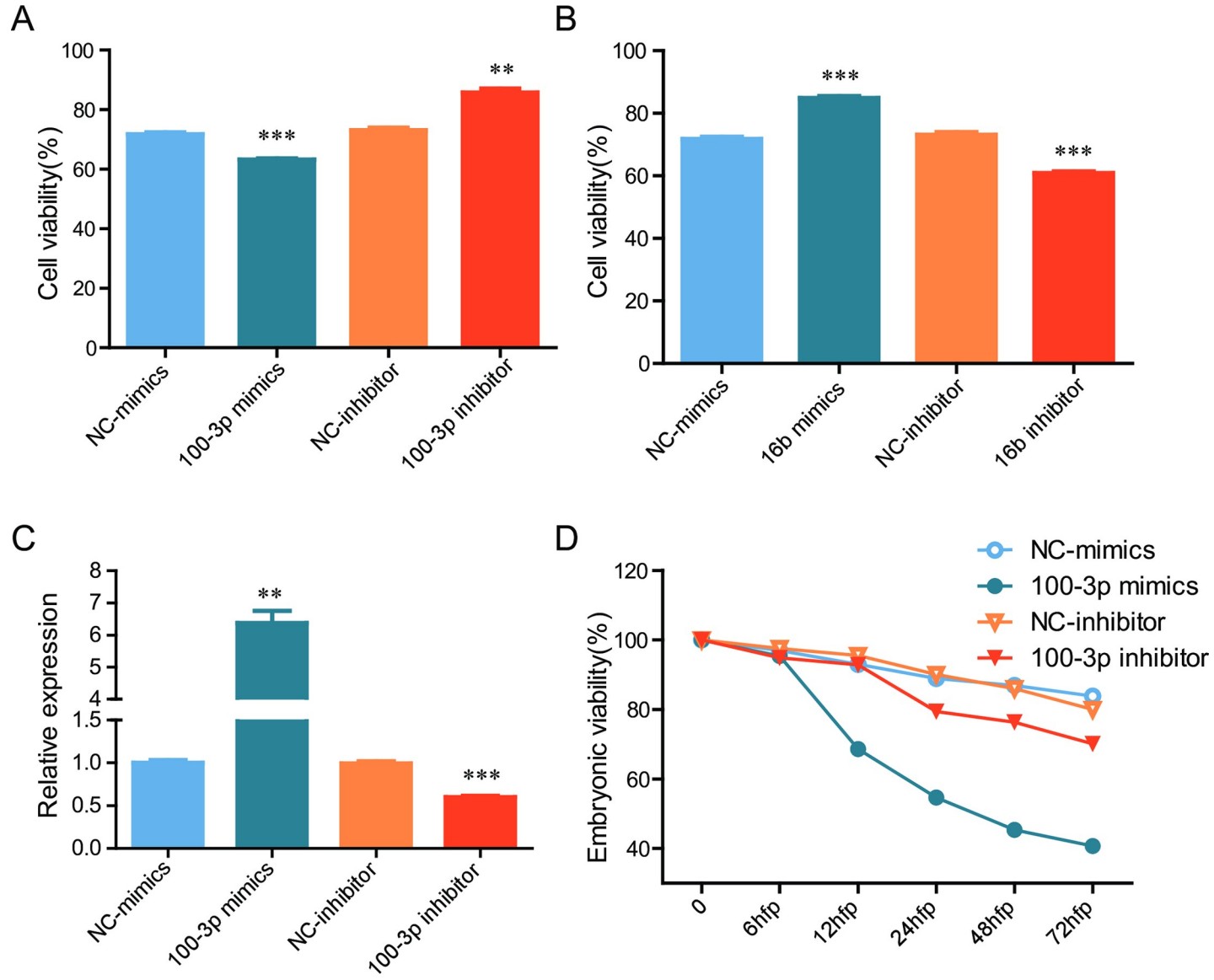

**Fig 6. dre-miR-100-3p and dre-miR-16b affect viability of ZF4 cells and embryonic development of zebrafish.** (A-B) ZF4 cells were transfected with indicated mimics or inhibitor, 24 hours later the cells were exposed to 10°C for 36 hours, then cell viability was examined. (C-D) Zebrafish embryos were microinjected with indicated mimics or inhibitor, 24 hours later the expression of miR-100-3p was detected by qRT-PCR (C), and viability of embryos was determined (n = 120) (D). *: p <0.05, **: p <0.01, ***: p <0.001. NC, negative control.

conditions [46]. miR-16 mimics can activate autophagy in non-small cell lung carcinoma cells [47]. miR-16-5p also shows protective roles in LPS-induced A549 cell injury [48]. However it has also been reported that miR-16 can induce apoptosis by targeting BCL2 [49]. In the present work, miR-16b decreased the expression of *rilp*, *atxn7*, *tnika*, *btbd9*, indicating these genes may contribute to miR-16b function. *rilp* has regulatory roles in cell motility affecting migration velocity, cell adhesion and cell spreading [50]. *atxn7* encodes a transcription factor related to chromatin remodeling [51]. The exact roles and mechanisms of dre-miR-16b in cold acclimation in fish remain unclear in this study. Overall, our results support that DE-miRNAs play important regulatory roles in cold acclimation in ZF4 cells.

## Conclusions

In this study, we systematically characterized the variation of miRNAs in zebrafish embryonic fibroblast ZF4 cells under cold acclimation conditions through high-throughput sequencing. There are 28 conserved and 19 novel miRNAs differentially expressed in cold acclimation in ZF4 cells. Those DE-miRNA are involved in regulation of phosphorylation, cell junction, intracellular signal transduction, ECM-receptor interaction and so on. Further study showed that dre-miR-16b and dre-miR-100-3p may contribute to cold acclimation through regulation of cell survival. The present data will help us understand the detailed mechanisms of miRNAs under cold pressure.

## Supporting information

**S1 Fig. Cluster analysis of novel DE-miRNAs.** A heatmap was generated based on fold change values of novel DE-miRNAs to visualize the expression patterns of the cold responsive miRNAs.
(TIF)

**S2 Fig. The binding sites of dre-miR-100-3p / dre-miR-16b and their target genes.**
(PDF)

**S3 Fig. The microscope photos of Trypan Blue staining of Fig 6A and 6B.**
(PDF)

**S1 Table. The sequences of primers for qRT-PCR and mimics/inhibitor.**
(XLSX)

**S2 Table. The known mature miRNAs.**
(XLSX)

**S3 Table. The mature novel miRNAs.**
(XLSX)

**S4 Table. The secondary structure of novel miRNAs.**
(PDF)

**S5 Table. The list of DE-miRNAs.**
(XLSX)

**S6 Table. The list of known DE-miRNAs target gene.**
(XLSX)

**S7 Table. GO and KEGG enrichment analyses of target gene.**
(XLSX)

## Acknowledgments

This study was supported by grants from National Natural Science Foundation of China (grant No. 81770165 awarded to BH and 31372516 awarded to JZ).

## Author Contributions

**Data curation:** Xiangqin Ji.

**Formal analysis:** Xiangqin Ji, Penglei Jiang.

**Funding acquisition:** Junfang Zhang, Bingshe Han.

**Investigation:** Xiangqin Ji, Penglei Jiang, Juntao Luo, Mengjia Li, Yajing Bai.

**Methodology:** Xiangqin Ji.

**Project administration:** Junfang Zhang, Bingshe Han.

**Resources:** Junfang Zhang, Bingshe Han.

**Software:** Xiangqin Ji.

**Supervision:** Junfang Zhang, Bingshe Han.

**Writing – original draft:** Xiangqin Ji.

**Writing – review & editing:** Xiangqin Ji, Junfang Zhang, Bingshe Han.

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
