## [Decision Letter · Decision Letter 0]

19 Aug 2019

PONE-D-19-20835

Identification and characterization of miRNAs involved in cold acclimation of zebrafish ZF4 cells

PLOS ONE

Dear Dr.Han,

Thank you for submitting your manuscript to PLOS ONE. After careful consideration, we feel that it has merit but does not fully meet PLOS ONE’s publication criteria as it currently stands. Therefore, we invite you to submit a revised version of the manuscript that addresses the points raised during the review process.

Reviewer #1: The authors characterized the miRNA profiles in ZF4 cells acclimated to mild cold stress for 30 days and the time-matched controls maintained at 28 oC. They identified cold acclimation-associated miRNAs and provided evidence that some of these differentially expressed miRNAs were involved in cold resistance. The bioinformatic analysis pipeline is reasonable and the data is well organized. The following concerns should be addressed before publication.

1) In the abstract, it should be made clear that the list of putative target genes for the DE-miRNAs was used for GO and KEGG enrichment analyses.

2) Why did the authors select foldchange > 2.5 as the threshold for DE-miRNA?

3) Line 144, the title of Table 1 should be “Primer sequences ……”

4) In the miRNA transfection and cell viability assays, what are the sequence and quantity of miRNA mimics and inhibitors used for transfection? Only concentration of these stuffs was provided. How about the transfection efficiency? The authors analyzed relative expression of these molecules after transfection. However, the efficiency of transfection, namely the proportion of cells that express the exogenous molecules after transfection is very important for understanding the data of function investigation.

5) Line 164, how the reads without sRNA sequences, ranging from 18 to 35 nt in length, were identified?

6) The level of representative targets for the critical miRNAs should be analyzed to further support the conclusion that they are involved in cold acclimation by regulating the abundance or translation of their cognate targets.

Reviewer #2: In this manuscript, the authors used RNA-seq to identify the miRNA roles in cold acclimation of ZF4 cell. Their previous study showed that DNA methylation and long non-coding RNAs (lncRNAs) are involved cold acclimation of zebrafish ZF4 cells when experimentally acclimated at 18˚C for 30 days, indicating that the authors have investigated the gene expression variation by RNA-seq methods using the ZF4 cell samples under two different temperatures. In this study,  RNA-seq data analysis has revealed differential expression of genes  including lncRNAs, microRNAs during cold acclimated at 18℃ for 30 days. Some concerns need to be addressed:

1) For the cold treatment, the ZF4 cell was seeded at 50% confluence and placed in an incubator at 18℃ and 5% CO2 for 30 days. Why di d you choose 50% confluence and how about the survival rates of cells after treatment for 30 days under low temperature?

2)To verify the miRNA function under cold pressure in ZF4 cells ,why did you choose to treat cells at 10℃,but not at 18℃，since the differential expression of miRNA  was identified in samples at 18℃ and 28℃？Moreove,  the measurement of cell viability  with Trypan Blue staining under microscope need be provided and the MTT experiments for the cell viability may be verified with Trypan Blue staining.

3)To comclude that the miRNAs are involved in fish cold acclimation, the microRNA functions should be tested in zebrafish embryos or larvae to answer if these microRNAs affect the early development of zebrafish or play a role in cold acclimation.

We would appreciate receiving your revised manuscript by Oct 03 2019 11:59PM. To enhance the reproducibility of your results, we recommend that if applicable you deposit your laboratory protocols in protocols.io, where a protocol can be assigned its own identifier (DOI) such that it can be cited independently in the future. For instructions see: http://journals.plos.org/plosone/s/submission-guidelines#loc-laboratory-protocols

We look forward to receiving your revised manuscript.

Kind regards,

Zongbin Cui, Ph.D.

Academic Editor

PLOS ONE

Journal Requirements:

2. We note that you are reporting an analysis of a microarray, next-generation sequencing, or deep sequencing data set. PLOS requires that authors comply with field-specific standards for preparation, recording, and deposition of data in repositories appropriate to their field. Please upload these data to a stable, public repository (such as ArrayExpress, Gene Expression Omnibus (GEO), DNA Data Bank of Japan (DDBJ), NCBI GenBank, NCBI Sequence Read Archive, or EMBL Nucleotide Sequence Database (ENA)). In your revised cover letter, please provide the relevant accession numbers that may be used to access these data. For a full list of recommended repositories, see http://journals.plos.org/plosone/s/data-availability#loc-omics or http://journals.plos.org/plosone/s/data-availability#loc-sequencing.

---

## [Author Response · Author response to Decision Letter 0]

2 Oct 2019

List of Responses

Dear Editors and Reviewers:

Thanks for your comments concerning our manuscript entitled "Identification and characterization of miRNAs involved in cold acclimation of zebrafish ZF4 cells" (Manuscript Number: PONE-D-19-20835). Those comments are very helpful for improving our data and we have made revision. The major corrections and the responds to the reviewer are as following:

Reviewer #1: The authors characterized the miRNA profiles in ZF4 cells acclimated to mild cold stress for 30 days and the time-matched controls maintained at 28℃. They identified cold acclimation-associated miRNAs and provided evidence that some of these differentially expressed miRNAs were involved in cold resistance. The bioinformatic analysis pipeline is reasonable and the data is well organized. The following concerns should be addressed before publication.

Thank you for your time reviewing this manuscript. We have made an extensive modification in the revised manuscript. The changes to our manuscript are highlighted by using red colored text. We would be glad to respond to any further questions and comments.

1) In the abstract, it should be made clear that the list of putative target genes for the DE-miRNAs was used for GO and KEGG enrichment analyses.

Reply: Thanks for your suggestion, the proper information has been provided in the revised abstract.

2) Why did the authors select foldchange > 2.5 as the threshold for DE-miRNA?

Reply: It is true that many researchers select foldchange > 1.5 (1) , 2 (2) as the threshold for DE-miRNA. Here we screened 55 known DE-miRNAs with foldchange > 2, when 28 known DE-miRNAs obtained with foldchange > 2.5. So here we set a stricter standard to screen out the most relevant miRNAs involved in cold acclimatiom. In the revised version, we provide the information of all DE-miRNA with foldchange > 1.5 for the readers in the Supplementary Table S5.

3) Line 144, the title of Table 1 should be “Primer sequences ……”

Reply: Thank you very much for pointing out this error. We have corrected this issue through the whole revised manuscript.

4) In the miRNA transfection and cell viability assays, what are the sequence and quantity of miRNA mimics and inhibitors used for transfection? Only concentration of these stuffs was provided. How about the transfection efficiency? The authors analyzed relative expression of these molecules after transfection. However, the efficiency of transfection, namely the proportion of cells that express the exogenous molecules after transfection is very important for understanding the data of function investigation.

Reply: The sequences and quantity of miRNA mimics and inhibitors have been provided in the revised manuscript. The sequences of mimics and inhibitors are summarized in Supplementary Table S1. In this study, the transfection efficiency ranges from 75% to 85% based on our experiments using a fluorescence-labeled control miRNA.

5) Line 164, how the reads without sRNA sequences, ranging from 18 to 35 nt in length, were identified?

Reply: Reads without sRNA sequences, ranging from 18 to 35 nt in length, were filtered using cutadapt. We have provided the information in the revised manuscript.

6) The level of representative targets for the critical miRNAs should be analyzed to further support the conclusion that they are involved in cold acclimation by regulating the abundance or translation of their cognate targets.

Reply: Thanks for your suggestion. In the revised manuscript, we showed that after treatment of ZF4 cells with mimics/inhibitor of miR-100-3p or miR-16b. An inverse correlation between the expression profiles of miRNAs (dre-miR-100-3p, dre-miR-16b) and their target genes (atad5a, cyp2ae1, lamp1; rilp, atxn7, tnika, btbd9) was confirmed by RT-qPCR 24h after transfection. And this relationship is consistent with our previous RNA-Seq data. We have provided our new observations in Fig. 5 to support this point.

Reviewer #2: In this manuscript, the authors used RNA-seq to identify the miRNA roles in cold acclimation of ZF4 cell. Their previous study showed that DNA methylation and long non-coding RNAs (lncRNAs) are involved cold acclimation of zebrafish ZF4 cells when experimentally acclimated at 18˚C for 30 days, indicating that the authors have investigated the gene expression variation by RNA-seq methods using the ZF4 cell samples under two different temperatures. In this study, RNA-seq data analysis has revealed differential expression of genes including lncRNAs, microRNAs during cold acclimated at 18℃ for 30 days. Some concerns need to be addressed:

Thank you for your time reviewing this manuscript. We have made an extensive modification in the revised manuscript. The changes to our manuscript are highlighted by using red colored text. We would be glad to respond to any further questions and comments.

1) For the cold treatment, the ZF4 cell was seeded at 50% confluence and placed in an incubator at 18℃ and 5% CO2 for 30 days. Why did you choose 50% confluence and how about the survival rates of cells after treatment for 30 days under low temperature?

Reply: For cold acclimation of ZF4 cells, acute exposure to 18℃ of ZF4 cells with confluence less than 50% showed significant growth inhibition and cell death. So we chose staring cold treatment at 50% confluence to make sure that most cells survive the whole process. Growth inhibition and slight cell death could be observed only in the first two weeks. After 30 days of treatment, the survival rates of cells are close to ZF4 cells cultured at 28℃. Detailed information can be found S1 Fig in reference (3).

2)To verify the miRNA function under cold pressure in ZF4 cells ,why did you choose to treat cells at 10℃,but not at 18℃，since the differential expression of miRNA was identified in samples at 18℃ and 28℃？Moreove, the measurement of cell viability with Trypan Blue staining under microscope need be provided and the MTT experiments for the cell viability may be verified with Trypan Blue staining.

Reply: Thanks for your suggestion. 

In our study, ZF4 cells were cold acclimated at 18℃, then DE-miRNA were identified. Cold acclimated cells differ form control cells by showing enhanced cold tolerance under severe cold exposure (10℃ in this work). And ZF4 cells at 50% confluence only showed slight cell death at 18℃, so it is hard to see the role of DE-miRNA at 18℃. So we chose to treat cells at 10℃ to verify the miRNA function during cold acclimation in ZF4 cells.

We have provided the microscope photos of Trypan Blue staining in Supplementary Fig.3 according to your suggestion. For our MTT assay kit, the attached instruction does not suggest its application at 10℃, and we observed that its application at RT/25℃ was likely disturbed by the cold-rewarming process, so we hesitated to provide this information for now.

3) To comclude that the miRNAs are involved in fish cold acclimation, the microRNA functions should be tested in zebrafish embryos or larvae to answer if these microRNAs affect the early development of zebrafish or play a role in cold acclimation.

Reply: According your comments, we investigated the effect of one selected miRNA on the early development of zebrafish. Our data showed that microinjection of zebrafish zygotes with miR-100-3p mimics led to increased death from 6 hpf, while miR-100-3p inhibitor showed only slight effect on the survival and development of zebrafish zygotes. The data have been provided in Fig.6.

We will work on more miRNA including miR-16b in our further research. 

Yours sincerely,

Bingshe Han

Bingshe Han, PhD

Building A Room A117

College of Fisheries and Life Science,

Shanghai Ocean University

No.999 Hucheng Ring Road

Shanghai 201306, China

Tel:86-21-61900476/Fax:86-21-61900492

Email: bs-han @shou.edu.cn

Reference: 

1. Tang Z, Zhang L, Xu C, Yuan S, Zhang F, Zheng Y, et al. Uncovering small RNA-mediated responses to cold stress in a wheat thermosensitive genic male-sterile line by deep sequencing. Plant physiology. 2012;159(2):721-38.

2. Liu W, Cheng C, Chen F, Ni S, Lin Y, Lai Z. High-throughput sequencing of small RNAs revealed the diversified cold-responsive pathways during cold stress in the wild banana (Musa itinerans). BMC Plant Biol. 2018;18(1):308.

3. Han B, Li W, Chen Z, Xu Q, Luo J, Shi Y, et al. Variation of DNA Methylome of Zebrafish Cells under Cold Pressure. PloS one. 2016;11(8):e0160358.

---

## [Decision Letter · Decision Letter 1]

28 Nov 2019

PONE-D-19-20835R1

Identification and characterization of miRNAs involved in cold acclimation of zebrafish ZF4 cells

PLOS ONE

Dear Dr.Han,

Thank you for submitting your manuscript to PLOS ONE. After careful consideration, we feel that it has merit but does not fully meet PLOS ONE’s publication criteria as it currently stands. Therefore, we invite you to submit a revised version of the manuscript that addresses the points raised during the review process.

Reviewer 2: The authors didn't show any evidence of the role of microRNAs in fish larvae, so they shoud change the conclusion that miRNAs are closely involved in cold acclimation in fish in the abstract. Does the miR-16b inhibitor have  effects on fish cold acclimation?

We would appreciate receiving your revised manuscript by Jan 12 2020 11:59PM. To enhance the reproducibility of your results, we recommend that if applicable you deposit your laboratory protocols in protocols.io, where a protocol can be assigned its own identifier (DOI) such that it can be cited independently in the future. For instructions see: http://journals.plos.org/plosone/s/submission-guidelines#loc-laboratory-protocols

We look forward to receiving your revised manuscript.

Kind regards,

Zongbin Cui, Ph.D.

Academic Editor

PLOS ONE

---

## [Author Response · Author response to Decision Letter 1]

5 Dec 2019

List of Responses

Dear Editors and Reviewers:

Thanks for your comments concerning our manuscript entitled "Identification and characterization of miRNAs involved in cold acclimation of zebrafish ZF4 cells" (Manuscript Number: PONE-D-19-20835R1). These comments are very helpful for improving our data and we have made revision. We would be glad to respond to any further questions and comments. The major corrections and the responds to the reviewer are as following:

Reviewer 2: The authors didn't show any evidence of the role of microRNAs in fish larvae, so they should change the conclusion that miRNAs are closely involved in cold acclimation in fish in the abstract. Does the miR-16b inhibitor have effects on fish cold acclimation?

Reply: 

They should change the conclusion that miRNAs are closely involved in cold acclimation in fish in the abstract

Thank you very much for pointing out this issue. We have corrected this issue in the abstract. 

Does the miR-16b inhibitor have effects on fish cold acclimation?

It is a very good question. As you pointed out in your comment, for now we do not have the data about the effect of miR-16b inhibitor on fish larvae, so we can not show the effects of miR-16b inhibitor on fish cold acclimation directly. Our ongoing work shows that miR-16b inhibitor microinjection led to increased abnormal development of zebrafish, so those fish larvae are not appropriate for evaluation of the effect of miR-16b inhibitor on fish cold acclimation, because it is hard to attribute observed changes to miR-16b inhibitor or early development defects. Further study of the role of miR-16b in fish may need conditional knockdown of miRNA in zebrafish.

The present data suggested the role of miR-16b in cold acclimation of ZH4 cells based on that miR-16b mimics enhanced cell survival under cold stress when miR-16b inhibitor led to decreased cell viability (Fig. 6B). So we hypothesized that miR-16b inhibitor may have the similar effect on fish cold acclimation.

If you have any suggestion, we will be very glad to hear from you.

Yours sincerely,

Bingshe Han

Bingshe Han, PhD

Building A Room A117

College of Fisheries and Life Science,

Shanghai Ocean University

No.999 Hucheng Ring Road

Shanghai 201306, China

Tel:86-21-61900476/Fax:86-21-61900492

Email: bs-han @shou.edu.cn

---

## [Editor Report · Decision Letter 2]

10 Dec 2019

Identification and characterization of miRNAs involved in cold acclimation of zebrafish ZF4 cells

PONE-D-19-20835R2

Dear Dr. Han,

We are pleased to inform you that your manuscript has been judged scientifically suitable for publication and will be formally accepted for publication once it complies with all outstanding technical requirements.

With kind regards,

Zongbin Cui, Ph.D.

Academic Editor

PLOS ONE

---

## [Editor Report · Acceptance letter]

27 Dec 2019

PONE-D-19-20835R2 

Identification and characterization of miRNAs involved in cold acclimation of zebrafish ZF4 cells 

Dear Dr. Han:

I am pleased to inform you that your manuscript has been deemed suitable for publication in PLOS ONE. Congratulations! Your manuscript is now with our production department. 

With kind regards,

on behalf of

Dr. Zongbin Cui 

Academic Editor

PLOS ONE